# MicroRNAs in Several Cutaneous Autoimmune Diseases: Psoriasis, Cutaneous Lupus Erythematosus and Atopic Dermatitis

**DOI:** 10.3390/cells9122656

**Published:** 2020-12-10

**Authors:** Sandra Domingo, Cristina Solé, Teresa Moliné, Berta Ferrer, Josefina Cortés-Hernández

**Affiliations:** 1Rheumatology Research Group, Lupus Unit, Hospital Universitari Vall d’Hebron, Institut de Recerca (VHIR), Universitat Autònoma de Barcelona, 08035 Barcelona, Spain; sandra.domingo@vhir.org (S.D.); fina.cortes@vhir.org (J.C.-H.); 2Department of Pathology, Hospital Universitari Vall d’Hebron, Institut de Recerca (VHIR), Universitat Autònoma de Barcelona, 08035 Barcelona, Spain; teresa.moline@vhir.org (T.M.); ferrerfabrega@yahoo.es (B.F.)

**Keywords:** microRNAs, skin autoimmunity, nanoparticles, biomarkers, pathogenesis, psoriasis, atopic dermatitis, cutaneous lupus erythematosus

## Abstract

MicroRNAs (miRNAs) are endogenous small non-coding RNA molecules that regulate the gene expression at a post-transcriptional level and participate in maintaining the correct cell homeostasis and functioning. Different specific profiles have been identified in lesional skin from autoimmune cutaneous diseases, and their deregulation cause aberrant control of biological pathways, contributing to pathogenic conditions. Detailed knowledge of microRNA-affected pathways is of crucial importance for understating their role in skin autoimmune diseases. They may be promising therapeutic targets with novel clinical implications. They are not only present in skin tissue, but they have also been found in other biological fluids, such as serum, plasma and urine from patients, and therefore, they are potential biomarkers for the diagnosis, prognosis and response to treatment. In this review, we discuss the current understanding of the role of described miRNAs in several cutaneous autoimmune diseases: psoriasis (Ps, 33 miRNAs), cutaneous lupus erythematosus (CLE, 2 miRNAs) and atopic dermatitis (AD, 8 miRNAs). We highlight their role as crucial elements implicated in disease pathogenesis and their applicability as biomarkers and as a novel therapeutic approach in the management of skin inflammatory diseases.

## 1. Introduction

MicroRNAs, also known as miRs or miRNAs, are small, highly conserved, non-coding RNA sequences that range from 19 to 25 nucleotides [1]. In recent years, thousands of miRNAs have been discovered employing new advances in molecular biology and bioinformatics, achieving relevance in translational research. miRNA biogenesis has been broadly investigated to establish that most miRNAs are transcribed from DNA sequences in the nucleus by RNA polymerase II (Pol II). Drosha, a member of the RNase III family, with protein DiGeorge syndrome critical region gene 8 (DGCR8), constitute the microprocessor complex that cleaves the primary miRNAs to generate a 70-nucleotide sequence called miRNA precursor [2,3]. This is exported by exportin-5 to the cytoplasm and then processed by RNase III endonuclease dicer. After processing, the terminal loop is removed resulting in a miRNA duplex that will be incorporated into the argonaute (AGO) family of proteins. The directionality of the miRNA determines the name of the mature form. Both 5-p and 3-p strands can be loaded into the AGO proteins; however, the selection of the 5p or 3p is based on the thermodynamic stability at 5’ ends of the miRNA duplex or a 5’ U at nucleotide position 1. Usually, strands with lower 5’ stability or 5’ uracil are preferentially loaded into AGO and are named “guide strands”. The unloaded strand is called a “passenger strand”, and it is degraded. After the miRNA duplex is unwound, it is incorporated into the RNA-induced silencing complex (RISC), forming the minimal miRNA-induced silencing complex (miRISC), and then, the miRNA 20 nucleotide’s (nt’s) mature form is able to recognise and target complementary mRNA sequences (Figure 1).

MicroRNAs can modulate the gene expression at the same cell where they are being synthetised, or they can be secreted, enveloped in extracellular vesicles (EVs), transported from a parental cell to neighbouring cells and regulate important biological functions in the recipient cells [4]. Moreover, a single miRNA may have multiple target genes, and a single gene may be targeted by multiple miRNAs [5], making them a powerful system for modulating and adjusting the gene expression, as they approximately regulate around 60% of all the protein-coding genes [6].

miRNAs are involved in development, organogenesis, proliferation and apoptosis, among other cell processes [7,8]. Under normal physiological conditions, microRNAs are regulating correct cell functions. However, in disease, microRNAs may change, inducing an altered gene expression that leads into an aberrant phenotype [9]. When they are dysregulated, they may alter relevant cellular processes, favouring pathogenic conditions. On the other hand, they may also play protective roles by trying to re-establish cell homeostasis. A miRNA balance is key for the correct functioning of cell and tissue physiology.

## 2. Role of miRNAs in the Skin Pathogenesis of Cutaneous Immune Disorders

Skin is the largest organ in the human body, and its development and morphogenesis require a highly regulated and undisrupted miRNA profile. miRNAs’ role in skin physiology is well-known [10,11], as they are involved in epidermal and dermal proliferation, pigmentation, aging, wound healing, skin microbiome and skin immunity, among other processes [12]. Recent findings show that miRNAs have a role in skin carcinogenesis [13] and in the pathogenesis of chronic inflammatory skin diseases, presenting lesional specific miRNA expression profiles that differ from healthy skin [14,15,16]. A better understanding of the role of miRNAs in autoimmune cutaneous diseases will enhance our knowledge of skin disease pathology. In this section, the most important miRNAs associated with psoriasis, cutaneous lupus disease (CLE) and atopic dermatitis (AD) are described with special emphasis on their role in the disease pathogenesis.

### 2.1. Psoriasis

Psoriasis is the most prevalent chronic inflammatory skin disease, with an estimated prevalence in adults ranging from 0.91% to 8.5%, varying by country and ethnicity [17]. Genetic and environmental factors in connection with abnormal regulation of the immune system are thought to be involved in pathogenesis of the disease. It is characterised by hyperproliferation and altered differentiation of epidermal keratinocytes and leukocyte infiltration—predominantly, neutrophils, myeloid cells and T cells, causing the secretion of inflammatory mediators such as TNF-α, interferon-γ (IFN-γ), interleukin (IL)-1, IL-17 and IL-22, which contribute to psoriatic inflammation [18]. It has been identified that the IL-23/IL-17 axis is the primary signalling pathway, leading to characteristic molecular and cellular changes in psoriatic skin [18]. It is widely accepted that psoriasis is a consequence of an impaired crosstalk between the immune system and the structural cells of the skin. Several studies have been conducted to reveal the role of miRNAs in psoriasis (Table 1 and Figure 2), highlighting the value of miRNA analysis. The role of miR-203, miR-31, miR-146a, miR-155-5p and miR-21 are described below. 

The first study in 2007 that reported a distinctive skin miRNA signature in psoriasis was published by E Sonkoly et al. [14]. The study identified miR-203 as a keratinocyte-derived microRNA related to inflammation by targeting the *SOCS3* gene. After that, further studies have confirmed the direct targeting [19] and its role in the regulation of psoriatic cytokines such as *TNF-α*, *IL-24* and *IL-8* in keratinocytes [20,21]. Moreover, in vitro experiments showed that miR-203 expression is upregulated after IL-17 stimulation in HaCat cells and that miR-203 is involved in the activation of the *JAK2/STAT3* signalling pathway, which contributes to VEGF secretion and the perpetuation of pathological angiogenesis [19]. Recently, it has been described that miR-203 negatively regulates keratinocyte proliferation through the direct targeting of *NR1H3* and *PPARG* [22]. Therefore, in psoriasis, the data suggest that miR-203 may be involved in skin epidermal hyperplasia, inflammation and angiogenesis (Figure 2). 

MiR-31 is known to be involved in normal skin physiology by regulating keratinocyte growth and hair differentiation [23]. High miR-31 levels can be detected in blood and lesional psoriatic epidermis, and its pathogenic role is primarily based on NF-κB signalling alteration [24,25]. NF-κB is a crucial mediator in the pathogenesis of psoriasis and participates in inflammation, cell proliferation, differentiation and apoptosis. Serine/threonine kinase 40 (*STK40*), a negative regulator of NF-κB signalling, has been identified as a direct target for miR-31 [26]. The study demonstrated that miR-31 promotes NF-κB via *STK40* targeting and leads to the secretion of CXLC1, CXCL8, CXCL5 and IL-1β, which promote vascular endothelial cell activation and attract leukocytes via chemotaxis into the skin. Primary keratinocytes treated with TGFβ1, which is highly expressed in psoriatic skin, showed an upregulation of miR-31 [26]. This effect was also observed when keratinocytes were treated with psoriatic-relevant cytokines: IL-6, IL-22, interferon-γ (IFN-γ) and TNF-α [25], demonstrating its importance in epidermal inflammation. This miRNA is also involved in keratinocyte proliferation, as in vivo studies showed that miR-31 promotes epidermal hyperplasia via the direct targeting of *PPP6C*, a negative regulator of the G1-S phase progression in the cell cycle [25]. Endothelin-1, a peptide involved in cell proliferation and leukocyte chemotaxis, has been positively associated with high levels of miR-31 in blood [24]. MiR-31 may play a role in dermal mesenchymal stem cells (DMSCs) [27], as low levels in DMSCs of psoriasis patients versus healthy controls are found, but this needs further investigation. Taken together, miR-31 has a crucial role in psoriasis by promoting epidermal proliferation and inflammation in lesion sites. 

MiR-146a is overexpressed in lesional skin and peripheral blood mononuclear cells (PBMCs) from psoriasis patients [14,28]. It is known for its negative role in epidermal inflammation by targeting NF-κB mediators *IRAK1* and *CARD10* and chemotactic attractant *CCL5* [29,30,31]. Xia et al. [28] showed that high levels of miR-146a in the skin and in PBMCs of psoriasis patients positively correlate with IL-17 levels in the skin and serum, respectively. However, target gene *IRAK1* was downregulated in PBMCs but not in the skin, showing the asynchronous expression of target genes in local lesions and peripheral PBMCs. In vivo studies using mice models of Psoriasis showed that miR-146a inhibition promoted earlier psoriasis-like onset, epidermal hyperproliferation, IL-17 skin inflammation and IL-8 secretion with the increased infiltration of neutrophils at lesion sites. 

MiR-155-5p has been shown upregulated in blood and psoriatic lesional skin [32,33]. It is involved in the keratinocyte cell cycle, as in vitro studies showed that miR-155 inhibition decreases keratinocyte proliferation and increases the expression of *PTEN*, *PIP3*, *AKT*, *BAX* and *BCL2* apoptotic genes [34]. Another study supported this finding by showing that miR-155-5p overexpression impairs keratinocyte apoptosis possibly by targeting *CASP3*, a validated direct target of miRNA-155-5p [35]. This miRNA is also involved in inflammation, as keratinocytes treated with TNF-α upregulated its expression. Moreover, when cells were stimulated with LPS and overexpressed miR-155-5p, there was an increase of *TLR4*; NF-κB proteins together with the levels of secreted TNF-α and IL-18, IL-6 and IL-1β via inflammasome *NLRP3/CASP1* activation [36]. *CXCL8* is also upregulated in miR-155-5p-overexpressed keratinocytes via the *GATA3/IL37* axis [37]. Elevated miR-155 levels have also been observed in DMSCs [38]; however, further research is needed to establish its role. Overall, miR-155-5p is involved in keratinocyte proliferation, apoptosis and inflammation in psoriasis. 

Finally, epidermal cells and infiltrated T cells in psoriasis lesions have shown increased miR-21 expression [39]. In vitro, it is regulated by LncRNA *MEG3* and regulates keratinocyte proliferation via the direct targeting of *CASP8* [40]. It also promotes proliferation by regulating the *AKT/PI3K* and TGFβ signalling pathways [41,42]. Regarding its role in inflammation, UVB-exposure promoted miR-21-3p upregulation in keratinocytes. This upregulation led to the production of proinflammatory cytokines IL-6 and IL-1β and chemokines CCL5 and CXCL10 in keratinocytes [42]. The expression of miR-21 is increased in both TH1 and TH2 differentiated T cells after activation with anti-CD3 and anti-CD28, indicating that this miR is involved in T-cell activation regardless the T-cell subtype. Moreover, it has an antiapoptotic effect on the activated T cells [39]. 

Therefore, this miRNA can contribute to psoriasis pathogenesis by modulating the cell cycle and inflammation in keratinocytes and T cells. 

Twenty-seven further miRNAs have also been described in psoriasis pathogenesis. They are detailed in Table 1 [43,44,45,46,47,48,49,50,51,52,53,54,55,56,57,58,59,60,61,62,63,64,65,66,67,68,69,70].

### 2.2. Cutaneous Lupus Erythematosus (CLE)

Cutaneous lupus erythematosus (CLE) is an autoimmune chronic disease that includes a broad range of dermatologic manifestations. CLE is divided into several subtypes, but discoid lupus erythematosus (DLE) is consistently reported as the most common subtype, and this may be because, as a chronic disorder, it is easier to identify compared to the more evanescent and nonscarring acute cutaneous and subacute cutaneous forms (SCLE) [80]. The CLE overall prevalence is estimated to be around 73.24 per 100,000 according to several USA studies [81]. The pathogenesis of CLE is not completely understood. It seems to be multifactorial and involves genetic predisposition, environmental triggers and abnormalities in the immune response. Findings indicate that UVB may act as a trigger, promoting skin damage and keratinocyte apoptosis. There may be a defective apoptosis/cell clearance, and the immune system is activated against autoantigens.

CLE lesions share extensive lymphocytic infiltrates with a high predominance of CD4 T cells with an imbalance towards T-helper 1, cytotoxic CD8+ T cells, as well as interferon type 1 signature and proinflammatory cytokines, IL-1α, IL-1β, IL-8, TNF-α and IL-6 [82]. To date, we have published the only microRNA study in CLE—in particular, discoid lupus [16]. The study identified in DLE lesions a different microRNA signature (miR-31 and miR-485-3p) when compared to nonlesional sites. The relevant identified miRNAs and their potential role in CLE pathogenesis are detailed below.

MiR-31 was identified as a keratinocyte-derived miR located in the DLE lesional epidermis. It is involved in epidermal apoptosis by promoting the upregulation of apoptotic genes (*BIM*, *BAX*, *P53* and *CASP3*) when overexpressed. Moreover, as in previous reports, we also found that it enhances NF-κB activation and the secretion of inflammatory cytokines such as IL-1β, IL-12 and IL-8 in keratinocytes. The crosslink between keratinocytes and lymphocytes is of critical importance in cutaneous autoimmune diseases, and it was found that miR-31 promotes the attraction of neutrophils and intermediate monocytes; therefore, it enhances the recruitment of immune cells into the DLE lesion sites, perpetuating inflammation.

MiR-485-3p was found in the infiltrating lymphocytes and fibroblasts in DLE lesions. It is involved in the activation of CD4+ and CD8+ T cells and, also, in promoting fibrosis by enhancing fibrotic genes *SMAD3*, *COL3A1* and *TGFβR* in fibroblasts. This fibrosis may occur, as miR-485-3p may be targeting peroxisome *PPARGC1A*, which is known for exerting a protective function of fibrosis development [83] and was found downregulated in fibroblasts that overexpressed miR-485-3p. Studies showing the direct target of *PPARGC1A* by miR-485-3p support this finding [71].

### 2.3. Atopic Dermatitis (AD)

Atopic dermatitis is a complex, systemic inflammatory disorder associated with a variety of clinical features [72]. It is the most common chronic inflammatory skin disease, with a prevalence of 15–20% of children and 1–3% of adults worldwide. It has high heritability; occurs frequently with other atopic diseases (asthma, allergic rhinitis and food allergies) and its incidence has increased two to three-fold in recent years in industrialised countries [73]. 

AD is characterised by an epidermal barrier disruption, activation of a T-helper 2 response and alteration of the skin microbiome [72]. IgE and eosinophils are increased, which, in turn, are thought to boost inflammation and skin damage through the production of reactive oxygen species, inflammatory cytokines and the release of toxic granule proteins [74]. miRNA expression profiles in the skin lesions of AD patients have been determined by microarray. The elevated expression of let-7i, miR-24, miR-27a, miR-29a, miR-193a, miR-199a and miR-222 was reported [15]. Gu et al. also reported a multitude of dysregulated miRNAs (e.g., upregulation: miR-4270, miR-211, miR-4529-3p and miR-29b and downregulation: miR-184, miR-135a and miR-4454) in AD skin biopsies [76]. From the identified miRNAs, we describe below the functional role of some of the most relevant in the skin pathogenesis of AD.

MiR-155-5p in AD lesional skin is predominantly expressed in infiltrating immune cells. This miR plays a role in the regulation of allergen-induced inflammation by targeting *CTLA4*, a negative regulator of T-cell activation [15]. It affects T-cell proliferation and differentiation by shifting towards a TH17 response [84]. The expression of this miR has been analysed in different disease stages in an AD mice model, and it was found to be increased in the elicited phase of the disease compared to controls [85]. Increased levels of miR-155-5p have also been detected in vitro in HaCAT cells stimulated with TNF-α, and it promotes inflammation and epithelial tight junction (TJ) changes by the direct binding of *PKIA* [86]. Taken together, miR-155-5p promotes T-cell activation, epidermal inflammation and TJ disruption in AD.

Previous studies have demonstrated that miR 146a is involved in the inflammatory response of atopic dermatitis (AD). MiR-146a expression is increased in keratinocytes and the chronic lesional skin of patients with AD expression. MiR-146a may have an anti-inflammatory role, alleviating chronic skin inflammation in atopic dermatitis through the suppression of innate immune responses in keratinocytes. It inhibited the expression of numerous proinflammatory factors, including IFN-γ-inducible and AD-associated genes *CCL5*, *CCL8*, and ubiquitin D (*UBD*) in human primary keratinocytes stimulated with IFN-γ, TNF-α or IL-1β. Studies demonstrated that miR-146a-mediated suppression in allergic skin inflammation partially occurs through the direct targeting of the upstream NF-κB signal transducers caspase recruitment domain, containing protein 10 and IL-1 receptor-associated kinase 1. In addition, *CCL5* was identified as a novel, direct target of miR-146a [31]. It is worth mentioning that the upregulation of miR-10a-5p, miR-29b, miR-223 and miR-151a have also been described in the inflammatory response and keratinocyte apoptosis for AD patients [75,76,77,78] (Table 1). 

Finally, miR-143 has been found downregulated in the lesional skin from AD patients [79]. It targets IL-13 receptor alpha 1 (*IL13R*), modulating IL-13 activity. IL-13 is a cytokine involved in TH2 responses that is highly expressed in AD skin lesions. Therefore, miR-143 may contribute to AD pathogenesis by favouring TH2 responses.

## 3. Common Deregulated miRNAs in Skin Autoimmune Conditions

Skin lesion transcriptome studies in different autoimmune skin diseases have described unique expression signatures for specific diseases but have also established a common cross-disease gene set among inflammatory skin diseases [87]. The comparative transcriptomic analyses of atopic dermatitis and psoriasis conducted by Choy D.F et al. revealed a shared neutrophil-attracting profile, which may be due to the underlying commonalities in IL-17 signalling [88]. The comparison of DLE and psoriatic lesions revealed differential clustering upon dimensionality reduction, although a certain overlap was observed, pointing toward the existence of a common cross-disease profile. Through a gene set enrichment analysis, the differential T-cell polarisation toward Th17 in psoriasis and Th1 in DLE was supported [89]. 

Since miRNA studies have become of interest in inflammatory skin disease research; we sought to clarify common and unique molecular and pathophysiologic features in inflamed skin by comparing psoriasis, CLE and AD. The comparison showed miR-31 to be upregulated in both psoriasis and DLE lesional skin, suggesting a shared NF-κB signalling inflammation pathway, a dysregulated keratinocyte apoptosis process and epidermal hyperplasia. On the other hand, miR-155 and miR-146a were found to be upregulated in psoriasis and AD. It seems they may share a regulation of Th17 and the production of chemoattractant cytokines such us CXCL8 fundamental for these conditions. 

As miRNA expression profiles are tissue-specific and, in many cases, define the physiological nature of cells, the fact that they are commonly dysregulated among skin conditions (Figure 3) indicates that miRNAs may exert similar pathogenic roles, and dysregulated signalling pathways may be shared between skin conditions. Studies comparing the miRNA profile between autoimmune skin disorders may be of value to understand their pathogenesis and to promote novel therapies. 

## 4. miRNAs as Potential Biomarkers in Skin Inflammatory Diseases

Circulating miRNAs have been described as biomarkers, since they are found in different body fluids such as serum, plasma, urine, saliva, tears, amniotic and cerebrospinal fluid. Some of the innate properties of miRNAs make them highly attractive as potential biomarkers. They are accessible, stable and resistant to ribonuclease degradation and easily detected in small volumes of samples using standard RT-qPCR [90]. It is not yet clear their origin or function; however, changes in a circulating miRNA profile have correlated with a large number of medical conditions, like gastrointestinal and cardiovascular diseases and primary and metastatic cancers [91]. However, miRNAs with important immunological modulation effects in pathogenesis are not necessarily the best biomarkers; for example, miRNAs in circulation show limited or no correlation with miRNA expression in skin, a difference from other conditions. Moreover, whether the dysregulated miRNAs in blood are disease-specific or related to systemic inflammation is unclear. To date, despite several miRNAs being studied, none of them are used as biomarkers in routine clinical practice. 

In this section, we review potential miRNAs as biomarkers in skin inflammatory diseases for early diagnosis, assessment of disease severity/activity, treatment response monitoring and associated comorbidities. 

### 4.1. miRNAs as Diagnostic Biomarkers

Studies in pathogenesis and biomarker research are most developed in psoriasis. Since there is a specific miRNA expression profile different from other skin diseases and healthy controls, miRNAs may be used as diagnostic markers. MiR-223 and miR-143 were found to be significantly upregulated in PBMCs from patients with psoriasis. A ROC analysis showed that miR-223 and miR-143 have the potential to distinguish between psoriasis and healthy controls [92], suggesting that miR-223 and miR-143 may serve as novel diagnostic biomarkers for psoriasis. In the same way, high levels of miR-369-3p in the serum and skin were also distinctive in psoriasis patients compared with healthy controls [93]. Hair studies have shown miR-424 levels to be significantly upregulated in the hair shaft of only patients with psoriasis compared with normal controls and those with atopic dermatitis. A receiver–operator curve analysis of hair shaft miR-424 to distinguish psoriatic patients from normal subjects showed an area under the curve of 0.77 [94]. Hair root levels of miR-19a were also significantly upregulated only in psoriasis compared with normal controls. In a characteristic (ROC) curve analysis for hair root miR-19a, to distinguish psoriasis patients from normal subjects, the area under the curve (AUC) was 0.87 [95]. The results support a putative use as noninvasive diagnostic markers for psoriasis. Finally, another study found that, by real-time PCR study, levels of miR-125b, miR-146a, miR-203 and miR-205 in the serum were significantly decreased in patients with psoriasis compared with normal subjects [96]. 

Several studies have been performed analysing the microRNA profile in systemic lupus erythematosus patients but fewer in CLE. In SLE, miRNAs have been analysed in the serum, plasma and urine, and their relation has been established with several lupus manifestations, such as nephritis, oral ulcers and lupus anticoagulant, among others [97]. Concerning cutaneous lupus, only one study that included SCLE and DLE patients and healthy donors screened a selected panel of miRNAs related with inflammation and fibrosis in the serum [98]. The study showed that miR-150, miR-1246 and miR-21 are downregulated in both SCLE and DLE compared to healthy controls; therefore, these miRNAs could be of use for CLE diagnosis. Regarding differences between CLE subtypes, no DLE-specific miRNAs were discovered; however, low levels of miR-23b and miR-146 appear characteristic of SCLE [99]. 

While miRNAs have been extensively investigated as biomarkers in allergic inflammatory conditions like asthma and allergic rhinitis, only a few studies have been conducted in atopic dermatitis. Lv Y et al. [100] focused on paediatric AD and found that miR-203 and miR-483-5p were upregulated in the serum from children with AD and showed areas under the ROC curve (AUC) > 0.7. Surprisingly, miR-203 was also found differentially expressed in urine from these patients, but in this case, it was downregulated. Another study analysing miRNAs in the plasma of children with AD concluded that miR-194-5p was downregulated in AD when compared with the control group, suggesting that it may be a valuable biomarker for AD diagnosis [101]. MiR-155 plays an important role in AD pathogenesis. Its expression was analysed in peripheral CD4+ T cells, and they found that miR-155 was significantly elevated in AD patient CD4+ T cells compared with healthy subjects, indicating that it may be a useful biomarker of the disease as well [15]. 

### 4.2. miRNA as Disease Activity and Severity Biomarkers

The present data suggest that certain miRNAs could potentially serve as psoriasis activity markers. To date, the disease severity of psoriasis is assessed by a PASI score and BSA [102]. However, serum markers reflecting disease activity have not been of clinical use in psoriasis. MiR-223 and miR-143, as previously described, are increased in PBMCs from psoriasis patients and positively correlated with the PASI score and with an area under the ROC curve (AUC) > 0.8 [92]. MiR19a upregulated in hair roots inversely correlated with duration onset and first visit to the hospital [95]. High levels of miR-1266 in the serum [103], reduction of miR-126 and upregulation of miR-200c in plasma [104,105] and elevated miR-146a and miR-155 in PBMCs can also be indicators of psoriasis activity [33,103]. On the other hand, miR-99a in PBMCs are negatively correlated with disease severity [103]. Finally, the serum and skin miR-369-3p levels were detected, and their correlations with disease severity were confirmed [106], in which miR-369-3p levels in skin had a positive linear relationship with the PASI scores [93,106]. Conversely, low miR-369-3p and miR-135b levels in the skin have been associated with disease improvement and lower severity [93,107].

To date, the disease activity in CLE is assessed by the Cutaneous Lupus Erythematosus Disease Area and Severity Index (CLASI) [108]. Only serum miR-150 levels have been identified to be inversely correlated with a CLASI activity score in SCLE patients. Since this miRNA has been associated with dermal and renal fibrotic processes, we can also infer that it may be involved in the activation of inflammatory and profibrotic pathways [109,110]. Therefore, miR-150 may be a good candidate to assess disease severity in SCLE. Further analyses of miRNAs in the plasma or in other biological fluids may yield interesting biomarker targets in CLE. 

One study aimed to identify a prognostic miRNA signature in children with AD from serum and urine by a genome-wide miRNA profiling analysis. MiR-203 levels were significantly upregulated in the serum of children with AD compared with healthy controls, and they were significantly associated with increased sTNFRI and sTNFRII. However, miR-203 was markedly decreased in the urine of children with AD [100]. Therefore, miR-203 is proposed as a potential biomarker for the severity of inflammation in childhood AD. It is not clear if the data can be extrapolated to adults, since children with AD may have different miRNA expression profiles compared to their adult counterparts. Research is needed in order to establish biomarkers in the adult population, to be able to predict disease prognosis and, importantly, AD biomarkers may help in the rapid detection of the relapse phase of the disease and treatment outcomes.

### 4.3. miRNA Levels to Monitor Therapeutic Effects

Changes in miRNA expression following therapy have been studied in psoriasis. Pathological T cells and dendritic cells can trigger abnormal keratinocyte proliferation in psoriasis progression via many cytokines, especially TNF-α. Thus, TNF-α is essential for the pathogenesis of psoriasis. The anti-TNF-α biological drug etanercept significantly suppressed a panel of 38 miRNAs, and validated serum levels of miR-106b, miR-26b, miR-142-3p, miR-223 and miR-126 were significantly downregulated [111]. On the other hand, adalimumab increased miR-23b. These data indicate that changes in the miRNA level can reflect a previously unknown effect of anti-TNF-α therapy [112]. Interestingly, levels of those miRNAs were not altered when patients were treated with methotrexate. Additionally, miR-146a-5p in PBMCs has been described to correlate with clinical efficacy in psoriatic patients treated with anti-TNF-α adalimumab [113] and miR-125a levels in plasma increased in etanercept-treated responder patients [114]. 

There are no studies evaluating the changes in miRNA levels in response to therapies in CLE or AD. Prognostic or early response miRNAs in DLE may be useful to monitor the disease and avoid the development of irreversible fibrotic scarring lesions. 

### 4.4. Associated Comorbidity Biomarkers

Cardiovascular disease, obesity, diabetes and hypertension have been found at a higher prevalence in psoriasis patients compared to the general population, suggesting that, although psoriasis affects mainly skin, comorbid conditions are related with the underlying chronic systemic inflammation. Studies have been conducted to evaluate miRNAs as biomarkers of comorbidities in psoriasis. Garcia-Rodriguez S. et al. described the upregulation of miR-33 in the plasma of psoriatic patients when compared to controls and correlated with elevated insulin blood levels [115]. Serum levels of miR-126 are negatively associated with carotid intima-media thickness in psoriatic patients [104]. Plasma levels of miR-200c are upregulated in psoriasis and correlated with cardiovascular risk [105]. These results indicate miRNAs may be of value to assess possible comorbidities in psoriasis. 

In AD children, miR-483-5p expression in serum has been found as an independent marker of IgE levels. However, the upregulation of miR-483-5p has been significantly associated with the presence of other simultaneous atopic conditions, such as rhinitis and/or asthma, in comparison with healthy controls [100]. Therefore, miR-483-5p may reflect the multiorgan/tissue involvement of AD (Table 2).

## 5. Targeting miRNAs to Treat Skin Autoimmune Diseases

To date, there is no fully effective therapy for several skin autoimmune diseases, and the drugs used are not exempt from significant side effects. In addition, there are refractory cases that do not respond to conventional treatment, and their therapeutic options are limited. Most drugs may reverse local skin inflammation, like in DLE, but they do not avoid the progression of fibrosis or irreversible sequelae. Therefore, there is a need for novel specific and safe therapeutic agents to treat these chronic skin inflammatory diseases. miRNA therapeutics are the most recent of a range of RNA therapies that have emerged over the last 10–15 years [117]. 

There has been an increase of miRNA profiling studies in skin inflammatory diseases, leading to the identification of differentially expressed miRNAs, key in disease pathogenesis. Data supports the concept that miRNAs represent valid drug targets for treatment [118]. There are two strategies to use microRNAs as genetic modulators: miRNA inhibitors and miRNA mimics [119]. miRNA inhibitors or anti-miRNAs are chemically synthesised single-stranded nucleic acids designed to specifically bind to endogenous mature miRNA molecules. When a miRNA inhibitor is administered, there is a reduction of the targeted endogenous miRNA, and as a result, the interaction of the miRNA of interest with its targets is prevented. This approach is used for those aberrant expressed miRs that are upregulated in disease. Conversely, miRNAs that are downregulated in disease may be replaced transiently by using miRNA mimics. The miRNA mimics are chemically synthesised miRs that mimic endogenous miRNAs and are able to restore miRNA expression levels to normal. Therefore, when administered, they can modulate the gene expression correctly and achieve appropriate cell functioning. 

### 5.1. In Vivo Approaches of miRNA Therapy for Skin Autoimmune Diseases

In vivo studies have been conducted to evaluate miRs as potential therapeutic agents in skin disorders, mainly in psoriasis. MiR-21 expression is increased in epidermal lesions of patients with psoriasis, and this leads to reduced epidermal *TIMP3* (tissue inhibitor of matrix metalloproteinase 3) expression and the activation of *ADAM17* (tumour necrosis factor-α-converting enzyme), which induces TNF-ɑ shedding. The inhibition of miR-21 by locked nucleic acid (LNA)-modified anti-miR-21 compounds ameliorated disease, reducing hyperplasia in patient-derived psoriatic skin xenotransplants in mice and in a psoriasis-like mouse model, suggesting that anti-miR-21 is a promising therapy for psoriasis [120]. Similarly, in an imiquimod-induced psoriasis mouse model, the subcutaneous administration of anti-miR-31 decreased acanthosis, dermal cellular infiltration and epidermal thickness hyperplasia [25]. Treatment with mimic-340 alleviated the psoriasis severity in the same mouse model through the downregulation of cytokine *IL17A* [121]. In addition, an intradermal injection of synthetic miR-146 mimics efficiently inhibited the development of psoriasiform skin and reduced the epidermal thickening and the number of infiltrating neutrophils [122]. These results highlight the potential of miRNA mimics as a therapy to alleviate skin inflammation. The topical administration of nanocarrier miRNA-210 antisense ameliorated the psoriasis-like dermatitis in mice, providing a potentially effective topical drug for psoriasis [123]. The delivery of mimic-145-5p into the skin also decreased the epidermal hyperplasia and ameliorated psoriasis-like dermatitis in mice [60]. 

In AD mice models, the treatment with anti-miR-155-5p inhibitors clearly reduced the thinning of the epidermis and reduction of the inflammatory skin cell infiltrates accompanied by decreasing levels of Th2 cytokines (IL-4, IL-5, IL-9 and IL-13) [85]. The results suggest that antimir-155 therapy would help reducing AD-associated inflammation. The IL-32γ inhibition of miR-205 led to an inactivation of NF-κB in AD mice models, suggesting a promising therapy for further study [124]. 

Other miRs have been investigated as a therapy in skin diseases. miR-132 plays a role in the wound-healing processes, and when liposome-formulated miR-132 mimics were topically applied in human ex-vivo skin wounds, they promoted healing [125]. miR-203 also plays a role in scarring and anti-miR-203 treatment accelerated wound closure and reduced scar formation in vivo in mice skin wound models [126].

So far, no in vivo miRNA therapy studies have been performed in CLE. miRNAs involved in skin fibrotic processes may be potential targets in DLE in order to avoid scarring.

In vivo experiments have reported good results in Ps and AD, giving excellent expectative of the clinical applicability of miRNAs therapy for autoimmune skin diseases. However, experiments performed using animal models not always translating the same positive results in human trials is an important limitation. Currently, novel technologies are applied, such as 3D skin models (skin organoids) using patients samples to validate in vivo results before performing human clinical trials.

### 5.2. Clinical Trials Using miRNA as a Therapy for Treating Skin Diseases

Since the discovery of miRNAs in 1993, a number of preclinical studies have been conducted. Consequently, there has been an increase in the number of patient treatments over the last decade, and some of them have progressed translationally, and phase I and phase II clinical trials are currently ongoing. There are only two clinical trials involving the applicability of miR-29 mimics to treat lesional skin. MiR-29 is known to have an antifibrotic role, and it may be helpful for the treatment of fibrotic skin diseases. It is found at a lower level in cutaneous scars, keloids and in the lesional dermis of scleroderma patients, as compared with healthy controls [127]. A double-blinded, placebo-randomised, within-subject controlled clinical trial (ClinicalTrials.gov, ID NCT02603224) was performed with Remlarsen (miR-29 Mimic) administrated intradermally in healthy volunteers. The study showed a downregulation in collagen expression and a reduction in the development of fibroplasia in incisional skin wounds, accompanied by a downregulation of miR-29 target genes *COL1A1*, *COL1A2* and *COL3A1*, strongly involved with fibrosis [128]. A phase II trial (ClinicalTrials.gov, ID NCT03601052) is ongoing to study the efficacy and safety of Remlarsen, as well as its pharmacokinetics in patients with a history of keloid scars.

However, there are not any clinical trials going on for psoriasis, atopic dermatitis or cutaneous lupus. It would be necessary to perform complete clinical trials in order to guarantee the possible clinical implementation of miRNA therapy into skin diseases treatments.

### 5.3. Topical Nanodelivery of miRNA

Topical-based miRNA administration may be an attractive approach for applying miRNA therapy in skin diseases [129]. Off-target effects, dilution and toxicity, often associated with systemic administration, may be avoided when miRNA formulation is applied directly on the lesional skin area. The most significant limitation in the transdermal application of miRNAs is the skin barrier, since the natural function of the skin is to protect the body from unwanted effects from the environment [130]. The stratum corneum is the primary barrier to the percutaneous absorption of compounds, as well as to water loss, providing most of the epidermal barrier function. It is composed of nonviable keratinocyte squames embedded in a lipid-rich matrix, making it nonpermeable and impedes the absorption of hydrophilic and lipophilic substances greater than 500 Da [131]. In addition, inflamed skin can complicate their penetration further [132]. Administration of the naked miRNA modulator will result in a poor outcome due to inefficient tissue penetration and degradation. An option to overcome this is to introduce chemical modifications that enhance their stability and delivery by increasing their resistance to degradation by the nucleases that are present in the skin. Emerging approaches for conveying small interfering RNA (siRNA) into the epidermis have been developed in recent years, mostly focusing on nonviral vectors such us liposomal or elastic vesicles, metal, liquid crystalline nanoparticles or with a peptide enhancer [133]. These studies showed the scope for the topical nanodelivery of miRNA for skin treatments (Figure 4a).

To our knowledge, one manuscript described a type of liposomal vesicle as a nanodelivery for miRNAs in the treatment of psoriasis [134]. Liposomes are lipid-based carrier nanovehicles, stable with a high loading efficacy and low cytotoxicity. Liposome formulation implies the formation of amphiphilic phospholipid bilayers that entrap an aqueous core. In this work, Lambert et al. combined DOTAP and sodium cholate and cholesterol as a stabiliser with 30% ethanol to create surfactant-ethanol-cholesterol-osomes (SECosomes), a type of liposome with high penetration ability. This system transmitted siRNA into a skin-humanised mouse model of psoriasis to silence the expression of human beta-defensin 2 and, also, an antimicrobial peptide that is overexpressed in psoriatic skin [134]. Altering the cholesterol composition and replacing sodium cholate with 1,2-dioleoyl-sn-glycero-3-phosphoethanolamine (DOPE), they obtained a modified SECosome called DDC642 that was capable of delivering pre-miR-145 or anti-miR-203 oligonucleotides in melanocytes and keratinocytes, respectively, to modulate their target mRNA levels (Figure 4b). In addition, DDC642 complexes repress target genes in the epidermis of human 3D psoriasis skin models without targeting the dermis or circulatory system [134]. This is a proof-of-concept that elastic liposomes could be used as a topical delivery system for miRNA therapeutics in psoriasis.

## 6. Conclusions, Limitations and Future Perspectives

miRNA profiling studies have identified that skin immune diseases have specific miRNA signatures and that miRNAs are playing pathogenic roles when dysregulated. Studying the effects of these dysregulated miRNAs can enhance the understanding of the etiopathogenesis of psoriasis, cutaneous lupus and atopic dermatitis. Moreover, the fact that they also are dysregulated in circulation makes them candidate biomarkers for differential diagnosis, a disease severity/prognosis assessment, and they may also be able to monitor the treatment response, which is of crucial importance to patients that may be refractory to standard treatments. Since they play a role in pathogenesis, they have the potential to be used as therapeutic targets. In vivo studies support that miRNAs are able to ameliorate skin disorders after being topically or intradermally administered. Nanoparticle encapsulations may help in their delivery, degradation avoidance and, specifically, cell uptake. MicroRNA-based therapy may be able to treat refractory dermatologic immune conditions more effectively, avoiding current treatment side effects. Although many preclinical studies have been conducted exploring the role of miRNAs in autoimmune skin pathogenesis and their application as therapy, at the moment, there is only one miRNA at the clinical trial stage. Their easy degradation, possible off-target effects and toxicity need to be avoided to generate an effective, directed and safe microRNA-based therapy. Moreover, their uses as biomarkers in clinic are still conditioned by the fact that there may be variability in detection assays and nonstandardised normalisation and data statistical analysis. Therefore, further investigation needs to be performed to be able to understand their role in pathogenesis and demonstrate their application both as promising biomarkers and as an effective treatment for autoimmune dermatological conditions.

## Figures and Tables

**Figure 1 cells-09-02656-f001:**
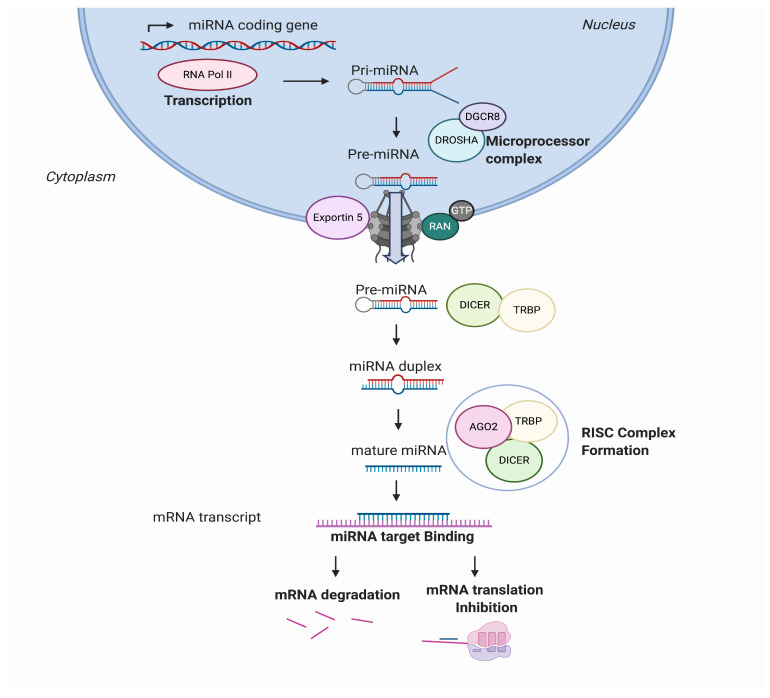
MicroRNA (miRNA) biogenesis and regulation of gene expression. miRNAs are transcribed from the genome into a pre-miRNA. The pre-miRNA is a smaller stem-looped structure that is transported from the nucleus to the cytoplasm by Exportin 5. Once in the cytoplasm, it is cleaved by DICER and TRBP and results into a small mRNA duplex that is around 20–25 nucleotides of length. The duplex is separated, and one of the strands is incorporated into the RISC, formed by AGO member proteins. The mature miRNA is then generated and binds specifically the mRNA transcript by complementary target recognition. The mRNA–miRNA union prevents the mRNA translation or leads into mRNA degradation and subsequent gene silencing. AGO: argonaute protein family and RISC: RNA-induced silencing complex.

**Figure 2 cells-09-02656-f002:**
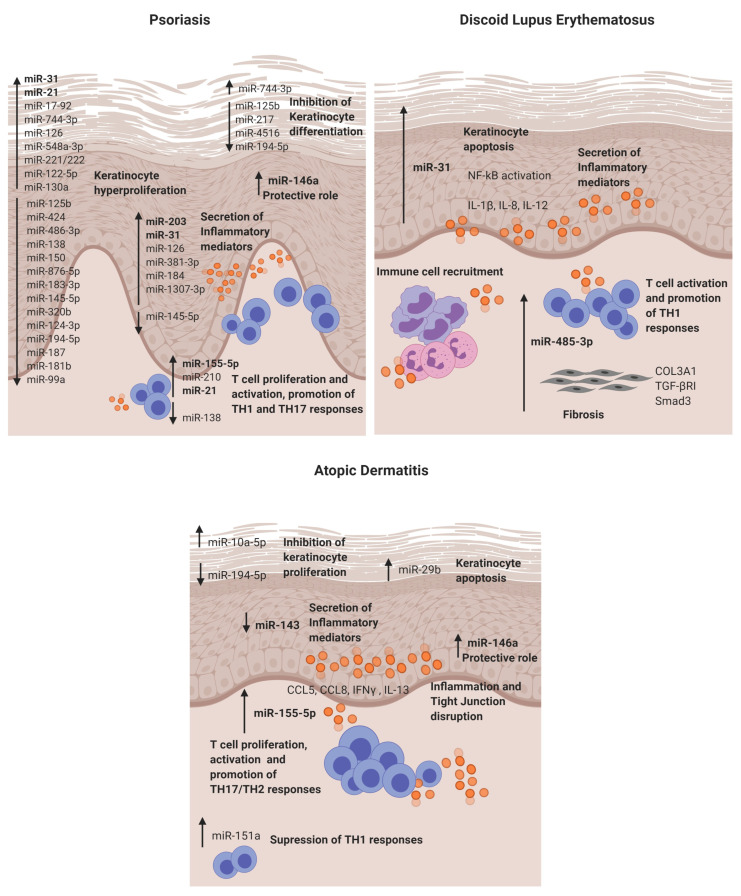
Dysregulated miRNAs involved in Psoriasis, discoid lupus erythematosus and atopic dermatitis and their roles in the disease pathogenesis. DLE: discoid lupus erythematosus and AD: atopic dermatitis.

**Figure 3 cells-09-02656-f003:**
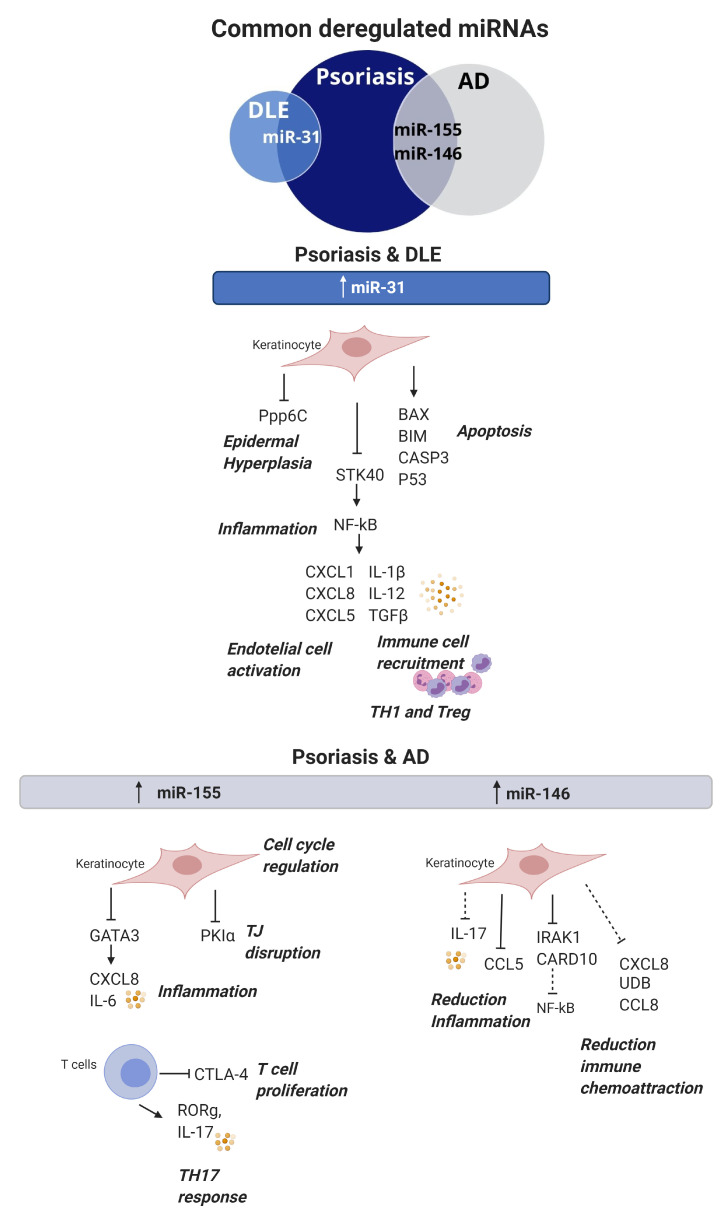
Commonly dysregulated miRNAs and their role in skin autoimmune diseases. MiR-31 is upregulated in both DLE and psoriasis; it participates in keratinocyte proliferation, apoptosis, inflammation and immune cell recruitment. MiR-155 and miR-146 are upregulated in both psoriasis and atopic dermatitis. MiR-155 is involved in keratinocyte proliferation, inflammation and TJ disruption, whereas, in T cells, it promotes proliferation and TH17 responses. MiR-146 has a protective effect and, when upregulated in keratinocytes, promotes a reduction of inflammation and immune chemoattraction. DLE: discoid lupus erythematosus, AD: atopic dermatitis and TJ: tight junction.

**Figure 4 cells-09-02656-f004:**
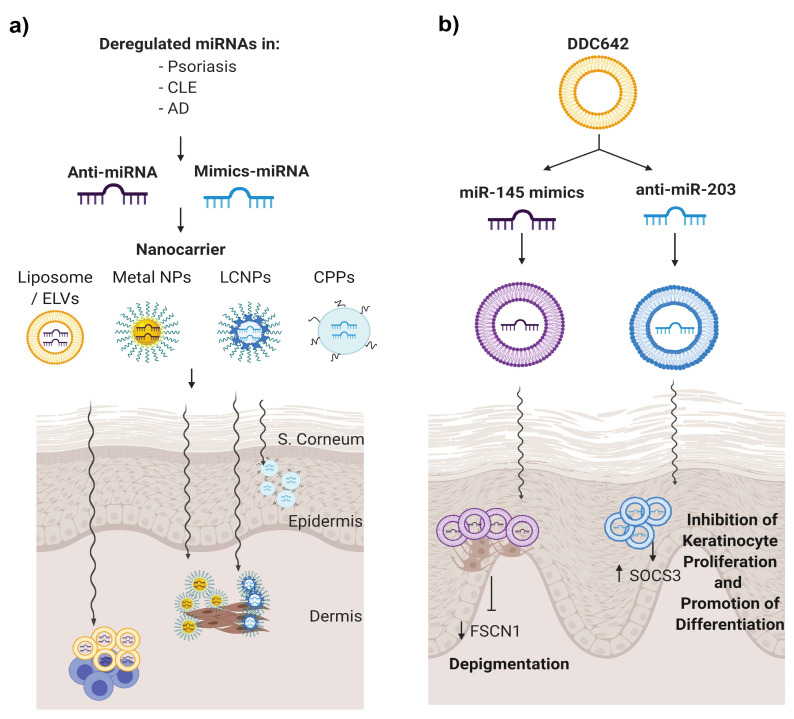
miRNA therapeutics for skin autoimmune diseases. (**a**) Dysregulated miRNAs in skin disorders may be potential candidates to be modulated to re-establish their expressions to normal conditions. Within this approach, anti-miRNA or mimics miRNA are encapsulated into nanovehicles to favour their penetration, avoid its degradation and be able to target the desired cells. (**b**) Deregulated miRNA in psoriasis miR-203 and miR-145 were selected, and therefore, anti-miR203 and miR-135 mimics were encapsulated in DDC624 liposome complexes, respectively, in a psoriasis tissue model.

**Table 1 cells-09-02656-t001:** Differentially expressed mRNAs in skin immune diseases. Tissue/cell/fluids in which microRNAs (miRNAs) are found dysregulated, miRNA expression, validated experimentally target genes, and their biological role in the skin are detailed.

miRNA	Condition	Tissue/Cell/Fluid	Expression	Target Genes	Biological Role	Ref.
**miR-203**	Psoriasis	Keratinocytes	Upregulated	*SOCS3* *NR1H3* *PPARG*	Keratinocyte proliferation, modulation of cytokines: TNF-α, IL-24 and IL-8 and angiogenesis.	[14,19,20,21,22]
**miR-31**	Psoriasis DLE	Keratinocytes Blood DMSCs	Upregulated (Blood and Keratinocytes) Downregulated (DMSCs)	*PPP6C* *STK40*	Keratinocyte proliferation and apoptosis. Promotes Inflammation via NFKB1 activation and chemokine and cytokine production (CXCL1, CXCL5, IL-8, IL-1B and IL-12). Neutrophile and intermediate monocyte recruitment.	[16,24,25,26]
**miR-146a**	Psoriasis AD	Keracinocytes Serum	Upregulated	*CCL5* *IRAK1* *CARD10*	Protective role disminishing keratinocyte proliferation and inflammation supressing IL-17, CCL5, CCL8 and IFNγ.	[14,28,29,30,31]
**miR-155**	Psoriasis AD	Keratinocytes Blood T cells	Upregulated	*CTLA4* *PKIA* *GATA3* *CASP3*	Promoted epidermal proliferation, inflammation, TJ disruption and inhibits apoptosis. T cell proliferation and promotion of TH17 responses.	[32,33,34,35,36,37,38,71,72,73,74]
**miR-21**	Psoriasis	Keratinocytes Blood T cells	Upregulated	*CASP8* *SMAD7*	T cell activation and inhibition of apoptosis. Keratinocyte proliferation and inflammation (IL-1β, CCL5 and CXCL10).	[39,40,41,42]
**miR-125b**	Psoriasis	Keratinocytes	Downregulated	*FGFR2*	Keratinocyte proliferation and differentiation.	[43]
**miR-424**	Psoriasis	Keratinocytes Serum	Downregulated	n.d.	Keratinocyte proliferation via MEK1/cyclin E1.	[44]
**miR-486-3p**	Psoriasis	Keratinocytes	Downegulated	*K17*	Keratinocyte proliferation.	[45]
**miR-138**	Psoriasis	Keratinocytes	Downregulated	*K17*	Keratinocyte proliferation and apoptosis reduction.	[46]
**miR-744-3p**	Psoriasis	Keratinocytes	Upregulated	*KLLN*	Keratinocyte proliferation and differentiation.	[47]
**miR-150**	Psoriasis	Keratinocytes	Downregulated	*HIF1A* *VEGFA*	Keratinocyte proliferation in hypoxic conditions.	[48]
**miR-876-5p**	Psoriasis	Skin Blood	Downregulated	ANG-1	HaCAT proliferation via PI3K/AKT, cell adhesion and invasion.	[49]
**miR-183-3p**	Psoriasis	Keratinocytes	Downregulated	*GAB1*	Proliferation and migration of HaCat cells.	[50]
**miR-548a-3p**	Psoriasis	Keratinocytes	Upregulated	*PPP3R1*	Keratinocyte proliferation.	[51]
**miR-217**	Psoriasis	Keratinocytes	Downregulated	*GFHL2*	Keratinocyte differentiation.	[52]
**miR-4516**	Psoriasis	Keratinocytes	Downregulation	*FN1* *ITGA9*	Accelerated migration, resistance to apoptosis and differentiation.	[53]
**miR-194-5p**	Psoriasis AD	Keratinocytes	Downregulated	*GRHL2* *HS3ST2*	Keratinocyte proliferation and inhibition of differentiation.	[54]
**miR-187**	Psoriasis	Keratinocytes	Downregulated	*CD276*	Keratinocute proliferation.	[55]
**miR-99a**	Psoriasis	Keratinocytes	Downregulated	*FZD5/FDZ8*	Keratinocyte proliferation.	[56]
**miR-130a**	Psoriasis	Keratinocytes	Upregulated	*STK40*	Apoptosis inhibition and cell viability and migration promotion. Direct regulation NFKB pathway via STK40 and inditect regulation of JNK/MAPK pathway via SOX9.	[57]
**miR-122-5p**	Psoriasis	Keratinocytes	Upregulated	*SPRY2*	Keratinocyte proliferation.	[58]
**miR-126**	Psoriasis	Keratinocytes	Upregulated	n.d.	Keraintocyte proliferation and inflammation increasing TNFa, IFNg, IL17A, IL-22 and decreasing IL-10. Apoptosis inhibition.	[59]
**miR-145-5p**	Psoriasis	Keratinocytes	Downregulated	*MLK3*	Cell proliferation and chemokine secretion via NF-kB and STAT 3 activation.	[60]
**miR-17-92**	Psoriasis	Keratinocytes	Upregulated	*CDKN2B*	Keratinocyte proliferation and immune chemotaxis via secretion CXCL9, CXCL10, supression of SOCS1 and STAT1 activation.	[61]
**miR-320b**	Psoriasis	Keratinocytes	Downregulation	*AKT3*	Keratinocyte proliferation and modulation of STAT3 and SAPK/JNKsingaling pathways.	[62]
**miR-124-3p**	Psoriasis	Keratinocytes	Downregulated	*FGFR2*	Keratinocyte prolfieration, migration and inflammation.	[63]
**miR-184**	Psoriasis	Keratinocytes	Upregulated	*AGO2*	Cytokine dependent depletion of AGO2.	[64]
**miR-221/222**	Psoriasis	Keratinocytes	Upregulated	n.d.	Keratinocyte and immune cells proliferation.	[65]
**miR-181-b**	Psoriasis	Keratinocytes	Downregulated	*TLR4*	Inflammation and keratinocyte proliferation.	[66]
**miR-1307-3p**	Psoriasis	Keratinocytes	Upregulated	n.d.	Induces inflammatory mediators IL-8, IL-6 and CCL20.	[67]
**miR-381-3p**	Psoriasis	Keratinocytes (EVs)	Upregulated	*FOXO1* *UBR5*	Crosslink with T cells inducing TH1/TH17 polarisation.	[68]
**miR-210**	Psoriasis	CD4^+^ T cells	Upregulated	*FOXP3*	Induces immune T cell dysfunction.	[69]
**miR-138**	Psoriasis	CD4^+^ T cells	Downregulated	*RUNX3*	Modulation of TH1/TH2 balance.	[70]
**miR-485-3p**	DLE	T cells Fibroblasts	Upregulated	*PPARGC1A*	T cell activation and promotion of fibrotic processes.	[16]
**miR-10a-5p**	AD	Keratinocytes	Upregulated	*HAS3*	Inhibitis keratinocyte proliferation.	[75]
**miR-29b**	AD	keratinocytes	Upregulated	*BCL2*	Keratinocyte apoptosis.	[76]
**miR-223**	AD	Blood	Upregulated	n.d.	Upregulation of HNMT indirectly to degrade excessive histamine.	[77]
**miR-151a**	AD	Blood	Upregulated	*IL12RB2*	Regulation of TH1 cytokines (IL-2, IFN γ).	[78]
**miR-143**	AD	Keratinocytes	Downregulatd	*IL13RA1*	Regulation of IL-13 activitu and TH2 inflammation.	[79]

AD, atopic dermatitis; DLE, discoid lupus erythematosus; DMSCs, dermal mesenchymal stem cells; Ref., Reference; n.d., not detailed and TJ, tight junction.

**Table 2 cells-09-02656-t002:** miRNAs as biomarkers and potential applications in autoimmune skin conditions (psoriasis, cutaneous lupus erythematosus and atopic dermatitis).

miRNA	Condition	Tissue/Cell/Fluid	Expression	Potential Application	Ref.
**miR-223** **miR-143**	Psoriasis	PBMCs	Upregulated	Diagnosis, assess disease severity and monitor treatment (metotrexate) response	[92]
**miR-424**	Psoriasis	Hair shaft	Upregulated	Diagnosis	[94]
**miR-19a**	Psoriasis	Hair root	Upregulated	Diagnosis and duration of disease	[95]
**miR-369-3p**	Psoriasis	Serum Skin	Upregulated	Diagnosis (skin and serum) and severity of disease (skin)	[93,106]
**miR-1266**	Psoriasis	Serum	Upregulated	Diasease activity	[116]
**miR-126**	Psoriasis	Plasma	Downregulated	Disease Severity Comorbidities (carotid thickness)	[104]
**miR-200c**	Psoriasis	Plasma	Upregulated	Diseaase activity and Comorbidities (cardiovascular disease)	[105]
**miR-155**	Psoriasis	PBMCs	Upregulated	Disease activity	[33]
**miR-146a**	Psoriasis	PBMCs	Upregulated	Disease activity Monitor treatment response (adalimumab)	[96,103]
**miR-99a**	Psoriasis	PBMCs	Downregulated	Disease activity	[103]
**miR-135b**	Psoriasis	Skin	Upregulated	Disease improvement	[107]
**miR-125a**	Psoriasis	Plasma	Downregulated	Diagnosis, and Monitor treatment response (etanercept)	[114]
**miR-33**	Psoriasis	Plasma	Upregulated	Detection of comorbidities (elevated insulin levels)	[115]
**miR-150** **miR-1246 miR-21**	CLE (SCLE and DLE)	Serum	Downregulation	Diagnosis	[98]
**miR-23b miR1246 miR-146**	SCLE	Serum	Downregulated	Diagnosis	[99]
**miR-150**	SCLE	Serum	Downregulated	Disease Seveity	[109,110]
**miR-203**	AD in children	Serum Urine	Upregulated in serum Downregulated in urine	Diagnosis and Disease severity	[100]
**miR-483-5p**	AD in children	Serum	Upregulated	Diagnosis and Detection of comorbidities (asthma/rhinitis)	[100]
**miR-194-5p**	AD in children	Plasma	Downregulated	Diagnosis	[101]
**miR-155**	AD	CD4^+^ T cells	Upregulated	Diagnosis	[15]

CLE, cutaneous lupus erythematosus; DLE, discoid lupus erythematosus; SCLE, subacute cutaneous lupus erythematosus; AD, atopic dermatitis and PBMCs, peripheral blood mononuclear cells.

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
