# Peer review of "MicroRNAs in Several Cutaneous Autoimmune Diseases: Psoriasis, Cutaneous Lupus Erythematosus and Atopic Dermatitis"

_cells, 2020, doi:10.3390/cells9122656_

Round 1
Reviewer 1 Report
Domingo et al. undertook a tremendous attempt to collect and review scientific data regarding the presence and role of microRNAs in cutaneous autoimmune diseases. And, in my opinion, they succeeded.
I really enjoyed reading their manuscript, as it is well written and compacted with vital epigenetic data. I find no major issues to be concerned. As to the minor ones – they are rather of editorial type, manageable in the editorial correction process.
Author Response
We appreciate your comments. We are going to correct all minor revisions during editorial process.

Reviewer 2 Report
This is a very interesting review about the different roles that play the miRNAs in 3 very common and importante skin diseases, related to their frequency and types associated with each one and describing the types existing, their possible role in the pathogenesis , diagnosis and treatment of these diseases
This review is very complete and well written containing a lot of information about the different aspects previously described, vary precised and with excellent illustrations in several Tables and Figures
The authors have performed an excellent job putting all the aspects on day and clearly argumented on all the aspects analyzed
The references are good enough, well selected and very useful

Author Response
This is a very interesting review about the different roles that play the miRNAs in 3 very common and important skin diseases, related to their frequency and types associated with each one and describing the types existing, their possible role in the pathogenesis, diagnosis and treatment of these diseases
This review is very complete and well written containing a lot of information about the different aspects previously described, vary precised and with excellent illustrations in several Tables and Figures
The authors have performed an excellent job putting all the aspects on day and clearly argumented on all the aspects analyzed
The references are good enough, well selected and very useful
Q1 : Title
I suggest to include “several” before cutaneous autoimmune diseases
We have added as your suggestion.
Q2 : Abstract and Keywords
Line 14 : Start with capital letter “MicroRNAs” at the beginning and add the acronym into parenthesis (miRNAs)
We have corrected as your suggestion.
Line 16 : Change “miRNA” by “Different”
We have changed.
Line 24 : You can include after Psoriasis his acronym (Ps) if you consider appropriated
Keywords seems too much . Can you remove someone?
We have added.
Q3 : 1. Introduction
Very well the MiRNA explained biogenesis and described and completed with an excellent drawing scheme, included at Figure 1
- Role of miRNAs in the skin pathogenesis…..
Good explained and summarized at Figure 2 and Table 1
2.1. Psoriasis 2
Line 87. You must precise if these percentages are in general population (including children and adults)
The percentage is from adult population, we have added it.
Line 98. You can add at 2007, after first study
We have added.
In the text included in Figure 2 at the bottom of Psoriasis Section you can make the correction of “T cell proliferation” instead of “proliertation”
We have corrected in the Figure 2.
The introduction of some small comment about the role of IL-23 would be also welcomed
We have added the role of IL-23 and IL-17 (Page 3, lines 95-96).
2.2. Cutaneous Lupus Erythematosus (CLE)
Line 180 : You must include the name of the name of the miRNA discovered , I suppose to be the MIR-31, because is an important finding you must emphasize it
We have added miR-31 and miR-485-3p.
2.3. Atopic Dermatitis (AD)
Very well explained and documented in Table 1 and Fig 2
- Common dysregulated miRNAs in skin AI conditions
It is very interesting to find these common associations of miRNAs between these different skin conditions
- miRNAs as potential biomarkers in skin inflammatory diseases 3
None of them are used as biomarkers in routine clinical practice. This is the main conclusion of this section
4.1. miRNAs as diagnostic biomarkers
It looks very interesting to use, especially in Psoriasis, more than in CLE or AD with a less defined pattern
4.2. miRNA as disease activity and severity biomarker
Is well explained and described
4.3 miRNA levels to monitor therapeutic effects
Positive findings in psoriasis, and no studies in the other two conditions
4.4. Associated co-morbidity Biomarkers
It is very relevant point to consider in the interpretation of the results found
- Targeting miRNAs to treat skin autoimmune diseases
The differentiation between the two modalities of miRNA (inhibition and mimics) are clearly exposed
5.1. In vivo approaches of miRNA therapy for these diseases
All of these studies are experimental in animals 4
5.2. Clinical trials using miRNAs as therapy for treating skin diseases
No CTS disposables in these 3 cutaneous diseases
5.3. Topical nanodelivery of miRNA
This may be a potential way to permit the transdermal application of chemical modifications to permit the access to the lesions of the viable product. Figure 4 is very illustrative and clear.
- Conclusions, limitations and future perspectives.
Further investigations needs to be performed before their clinical application arrive in an effective and sure way
7.- References
The authors have selected a good number of references (138) with a very high high scientific content and all of them are very relevant
We appreciate your positive inputs.

Reviewer 3 Report
The manuscript „MicroRNAs in cutaneous autoimmune diseases: Psoriasis, Cutaneous Lupus Erythematosus and Atopic Dermatitis“ covers the topic of the pathogenetic pathways of few skin diseases and the role of microRNA in them.
I consider this research very valuable, welcomed and complex – it's one of the very important reviews of these new factors which participate in many diseases. As it is a somewhat specific molecular article, I could comment only on the clinical part of the manuscript.
Abstract: It will be useful to add some numerical data on some of the previously analyzed data and factors.
Also, I suggest using the same term for specific type of lupus – there are different terms and sometimes it is not clear the real meaning.
The terms in vitro and in vivo should be written italic.
What is the possibility of further clinical use of the research results for patients in general and for the management of their treatment? Can you elaborate on this with more details?
Also, cited references are sometimes written in various styles (somewhere eg. ref. 10 there is 3 authors et al., while somewhere such as ref. 15 there is10 authors et al.) – they should be written according to the same journal rules for the authors.
Also, the manuscript needs to be proofread by a native English speaker or an English language specialist. There are some type fellers.
Thus, it is an especially important manuscript as there are a small number of articles concerning the subject. There are very nice figures and tables, which are very useful for further authors and general knowledge on this subject.
After all of the mentioned corrections are made, according to my opinion, this article could be published as REVIEW report.
(MINOR REVISIONS ARE NEEDED)
Author Response
The manuscript „MicroRNAs in cutaneous autoimmune diseases: Psoriasis, Cutaneous Lupus Erythematosus and Atopic Dermatitis“ covers the topic of the pathogenetic pathways of few skin diseases and the role of microRNA in them.
I consider this research very valuable, welcomed and complex – it's one of the very important reviews of these new factors which participate in many diseases. As it is a somewhat specific molecular article, I could comment only on the clinical part of the manuscript.
Abstract: It will be useful to add some numerical data on some of the previously analyzed data and factors.
We have included some numerical data in the abstract following your suggestion.
Also, I suggest using the same term for specific type of lupus – there are different terms and sometimes it is not clear the real meaning.
We have used three terms for lupus: Systemic Lupus Erythematosus (SLE), Cutaneous Lupus Erythematosus (CLE) and Discoid Lupus Erythematosus (DLE). We consider not use only a unique term to describe lupus because they are different between them.
SLE include all systemic manifestation of lupus (renal, cutaneous, articular, etc); CLE are lupus with cutaneous manifestations; DLE is a type of CLE with a specific lesions that are red, inflamed and with scaly and crusty appearance.
The terms in vitro and in vivo should be written italic.
We have corrected as your suggestion.
What is the possibility of further clinical use of the research results for patients in general and for the management of their treatment? Can you elaborate on this with more details?
We have included this concept in section 5 “Targeting miRNAs to treat skin autoimmune diseases” (Page 8, lines 472-477 and lines 494-496 ).
Also, cited references are sometimes written in various styles (somewhere eg. ref. 10 there is 3 authors et al., while somewhere such as ref. 15 there is10 authors et al.) – they should be written according to the same journal rules for the authors.
We have corrected the references style following the journal instructions.
Also, the manuscript needs to be proofread by a native English speaker or an English language specialist. There are some type fellers.
Manuscript has been revised for English language specialist from Vall Hebron Institut of Research.
Thus, it is an especially important manuscript as there are a small number of articles concerning the subject. There are very nice figures and tables, which are very useful for further authors and general knowledge on this subject. After all of the mentioned corrections are made, according to my opinion, this article could be published as REVIEW report.
We appreciate your positive inputs.
